# Association of Homocysteine, S-Adenosylhomocysteine and S-Adenosylmethionine with Cardiovascular Events in Chronic Kidney Disease

**DOI:** 10.3390/nu17040626

**Published:** 2025-02-10

**Authors:** Insa E. Emrich, Rima Obeid, Jürgen Geisel, Danilo Fliser, Michael Böhm, Gunnar H. Heine, Adam M. Zawada

**Affiliations:** 1Faculty of Medicine, Saarland University, 66421 Homburg, Germany; rima.obeid@uks.eu (R.O.); danilo.fliser@uks.eu (D.F.);; 2Department of Internal Medicine III, Saarland University Medical Center, 66421 Homburg, Germany; 3Clinical Chemistry and Laboratory Medicine/Central Laboratory, Saarland University Medical Center, 66421 Homburg, Germany; 4Department of Internal Medicine IV, Saarland University Medical Center, 66421 Homburg, Germany; 5AGAPLESION Markus Krankenhaus, 60431 Frankfurt, Germany

**Keywords:** chronic kidney disease, cardiovascular events, one-carbon (C1) metabolism, homocysteine, S-adenosylhomocysteine, S-adenosylmethionine, cardiovascular risk

## Abstract

**Background:** Patients suffering from chronic kidney disease (CKD) have a high risk of premature cardiovascular morbidity and mortality. It has been suggested that elevated homocysteine (Hcy) or disturbances in the transmethylation pathway may contribute to this high cardiovascular risk burden due to epigenetic mechanisms. The objective of this study was to explore the prognostic value of Hcy, S-adenosylhomocysteine (SAH) and S-adenosylmethionine (SAM) (one-carbon (C1)-metabolites) among patients with CKD. **Methods:** Plasma concentrations of Hcy, SAM and SAH were measured among 297 participants with CKD (KDIGO GFR category G2–G5). The predefined endpoint was the occurrence of major cardiovascular events (MACE), defined as carotid, coronary and peripheral arterial revascularization, stroke, acute myocardial infarction, major amputation, cardiovascular death and all-cause mortality during a median (IQR) follow-up period of 4.0 [3.2; 4.3] years. **Results:** Among all participants, the median (IQR) of plasma Hcy, SAH, and SAM levels were 16.6 [13.5; 21.2] µmol/L, 41.5 [26.6; 63.9] nmol/L, 183.4 [151.1; 223.5] nmol/L, respectively. Estimated glomerular filtration rate (eGFR) correlated more strongly with plasma SAH (r = −0.588) than with SAM (r = −0.497) and Hcy (r = −0.424). During the follow-up period, 55 participants experienced MACE. In a univariate Kaplan Meier analysis, all three C1-metabolites were significantly associated with the occurrence of the primary outcome. In a Cox-regression analysis, the association between Hcy and MACE was not significant after adjustment for age and sex (hazard ratio (HR) and 95% confidence intervals (95% CI) for the 3rd vs. 1st tertile = 1.804 (0.868–3.974)). Both SAH and SAM were not associated with MACE after adjustment for age, sex and additionally for renal function markers (SAH: HR 3rd vs. 1st tertile 1.645 95% (0.654–4.411); SAM: HR 3rd vs. 1st tertile 1.920 95% CI (0.764–5.138)). **Conclusions**: In people with CKD, plasma Hcy, SAH and SAM were not independent predictors of MACE after adjustment for age, sex and renal function. Disturbed renal function may explain elevated C1-metabolites and disturbed transmethylation, while this pathway is not likely to be an appropriate access point to modify the risk of cardiovascular events in CKD patients.

## 1. Introduction

Chronic kidney disease (CKD) affects hundreds of millions of people worldwide and its prevalence is steadily increasing [1]. Traditional cardiovascular risk factors do not fully explain the high risk of premature cardiovascular morbidity and mortality among patients with CKD [2]. Identification of other (non-traditional) cardiovascular risk factors may offer potential for the prevention of cardiovascular diseases [3].

Hyperhomocysteinemia (defined as homocysteine (Hcy) concentrations above 12 to 15 µmol/L) has been discussed as an atherosclerosis risk factor or risk marker [4] and has been shown to be associated with cardiovascular events and mortality in people with CKD [5]. Disturbed one-carbon (C1) metabolism may affect epigenetic regulation processes, such as DNA methylation [6] by mechanisms related to imbalanced S-adenosylhomocysteine (SAH) and S-adenosylmethionine (SAM). SAM is a cofactor for numerous methyl transferases in the cell, whereas SAH represents an allosteric inhibitor that prevents binding of SAM to the enzymes. Plasma concentrations of SAH are markedly elevated in people with CKD, as the kidneys are the major site of SAH disposal [7]. In addition, SAH is irreversibly converted to Hcy, suggesting that SAH is a significant source of Hcy in people with CKD. Recent studies demonstrated that SAH, instead of Hcy, may be the real culprit in the pathogenesis of cardiovascular disease [8,9]. A first prospective study in people undergoing coronary angiography found that plasma SAH is an independent predictor of cardiovascular events [10]. No prospective data exist in people with CKD. The present study aimed at investigating the association between plasma concentrations of SAH, SAM and Hcy in CARE FOR HOMe FU (cardiovascular and renal outcome in CKD 2–4 Patients—The Fourth Homburg evaluation Follow Up), a longitudinal study among people with CKD KDIGO glomerular filtration rate (GFR) category G2–G4.

## 2. Materials and Methods

### 2.1. Study Population

The CARE FOR HOMe (cardiovascular and renal outcome in CKD 2-4 Patients—The Fourth Homburg evaluation) study recruited people with CKD KDIGO GFR category G2–G4 regularly visiting the outpatient clinic of the Department of Internal Medicine IV, Homburg, Germany. The clinical outcomes were assessed during annual follow-up visits.

The design and baseline characteristics of the participants have been published elsewhere [11]. Detailed inclusion and exclusion criteria are presented in the Appendix A.

The present analysis is based on visits between March 2012 and March 2013, where we collected blood samples for measurement of plasma concentrations of SAH, SAM and Hcy, renal function markers and those of routine laboratory parameters. We included all CARE FOR HOMe participants who showed up for the annual follow-up examination (CARE FOR HOMe FU study). Out of 444 participants who were originally recruited, 297 accepted to join this study and provided blood samples for the biomarkers. In contrast to the original baseline recruitment, we also included those CARE FOR HOMe participants who progressed to CKD KDIGO GFR category G5 between study initiation and the 2012/2013 visit.

The responsible ethical committee “Ethik-Kommission der Ärztekammer des Saarlandes” approved the study (follow-up study code 08/10), and all participants provided written informed consent for the study. The authors adhered to the Declaration of Helsinki.

### 2.2. Exposure Definition

The study exposures were plasma concentrations of Hcy, SAH and SAM that were measured in blood samples collected at the 2012/2013 study visit. For the present analysis, the 2012/2013 visit was the index date. The median (IQR) follow-up period from the 2012/2013 study visit was 4.0 [3.2; 4.3] years.

### 2.3. Outcome Assessment

The predefined endpoint was the occurrence of major cardiovascular events (MACE) defined as acute myocardial infarction, surgical or interventional coronary/cerebrovascular/peripheral arterial revascularization, stroke and major amputation, cardiovascular mortality and all-cause mortality.

People who were unable or unwilling to attend the annual visits at the study center were contacted by telephone to inquire about their health conditions. This information was validated by contacting the treating physicians and reviewing the medical records. Two medical practitioners verified all reported events; in case of disagreement, a third one was involved to make the final decision. The mortality events were verified through reviewing death certificates. The clinical investigators were blinded for the results of the blood analyses when they assessed the study outcome. No participant was lost to follow up from the 2012/2013 visit to the end of the follow-up period.

### 2.4. Baseline Characteristics

A standardized questionnaire was used to collect information on previous cardiovascular events, comorbidities such as hypertension and diabetes mellitus, family predisposition, medication use and smoking habits. People who had stopped smoking at least one month before study entry were considered as former smokers. The criteria to define diabetes mellitus were self-reported diabetes mellitus, a non-fasting blood sugar level ≥ 200 mg/dL, a fasting blood sugar level ≥ 126 mg/dL or current use of hypoglycemic medication. Body mass index (BMI) was calculated as individuals’ body weight divided by the square of their height [kg/m^2^].

Systolic blood pressure (SBP) and diastolic blood pressure (DBP) were measured after five minutes of rest by an automated blood pressure device (GE Carescape DINAMAP V100; GE Healthcare, Chicago, IL, USA). eGFR was estimated by the 2009 creatinine-based CKD-EPI equation [mL/min/1.73 m^2^] and albuminuria was determined by albumin–creatinine ratio [mg/g] in spot urine.

Following an overnight fast (8–10 h), venous blood samples were collected using dry tubes and EDTA-containing tubes. Blood parameters were measured at the Central Laboratory of the Saarland University Medical Center. The EDTA blood-containing tube was immediately placed on ice to avoid an artificial increase in Hcy. The sample was centrifuged within 60 min for 10 min at 2000× *g,* and the EDTA plasma was separated from the red blood cells. An aliquot of 1 mL of the EDTA plasma was immediately acidified by adding 100 μL 1N acetic acid and frozen at −80 °C until measurement of SAM and SAH concentrations. The concentrations of SAM and SAH in the acidified plasma were measured using an established method on a UPLC-MS/MS system [12]. Another portion of the EDTA plasma sample was used to measure concentrations of Hcy within 24 h using a fluorescence polarization immunassay on the Abbott AxSYM analyzer [Abbott laboratories, North Carolina, IL, USA].

### 2.5. Statistical Analysis

Categorical variables are presented as percentages and compared by Fisher’s test and chi-square test. Continuous variables are shown as mean ± standard deviation (SD) (in case of normal distribution of the data) or as median (interquartile range (IQR)) for skewed variables. The one-way analysis of variance (ANOVA) test was used to compare continuous variables between two groups or more. The correlation coefficients between various continuous variables were calculated using the Spearman test. Univariate Kaplan–Meier analyses with consecutive log-rank testing for event-free survival were performed to study the association between the one-carbon metabolites and MACE. Additionally, Cox regression analyses were conducted and the hazard ratio, and 95% confidence intervals (HR (95% CI) for MACE were computed for the upper two tertiles of the plasma concentrations of the metabolites versus the lowest tertile (reference group). Collinearity between confounders were assessed by analyzing variance inflation factor (VIF). VIF > 10 was considered to indicate serious collinearity, thereby inducing unstable cox regression coefficients with large confidence intervals. Variables with high collinearity were excluded. Cox regression analyses were either crude (Model 1); adjusted for age and sex (Model 2); adjusted for age, sex, eGFR, and log-albuminuria concentrations (Model 3); or adjusted for the same variables in Model 3 plus current smoking, prevalent cardiovascular disease and known diabetes mellitus (Model 4). No imputation was performed for missing data of the covariates.

Data management and statistical analysis were performed with PASW Statistics 18.

## 3. Results

### 3.1. Participants’ Characteristics

Among the 297 participants, 117 (39.4%) were female, 32 (10.8%) were active smokers, 106 (35.7%) suffered from diabetes mellitus and 95 (32.0%) had prevalent cardiovascular disease. The median (IQR) age was 69.1 [60.4; 76.8] years. Mean ± SD of plasma creatinine concentrations was 1.64 ± 0.72 mg/dL and median (IQR) of eGFR was 45 [32; 57] mL/min/1.73 m^2^. Baseline characteristics of the study participants, according to the CKD KDIGO GFR category, are presented in Table 1. Most of the participants belong to CKD KDIGO GFR category G3 (92 participants in G3a and 79 participants in G3b), whereas 8 patients were on dialysis. Compared to people in early CKD stages, those classified to more advanced CKD stages were older, had a more frequent history of prevalent cardiovascular disease, lower diastolic blood pressure, higher concentrations of phosphate in plasma and albumin in urine, and lower concentrations of HDL cholesterol. The number of smokers was highest in patients with CKD KDIGO GFR category G2. Baseline characteristics differentiated by reaching MACE or not are presented in the Appendix A.

### 3.2. Parameters of One-Carbon Metabolism in Cross-Sectional Analyses

Among the total population, median (IQR) plasma concentrations of SAH, SAM and Hcy were 41.5 [26.6; 63.9] nmol/L, 183.4 [151.1; 223.5] nmol/L and 16.6 [13.5; 21.2] μmol/L, respectively. All three metabolites showed inverse correlations with eGFR (Table 2). The correlation between plasma concentrations of SAH and eGFR was strong (Spearman correlation coefficient r= −0.588; *p* < 0.001). The SAM/SAH ratio (often used as a surrogate marker of methylation capacity) showed a direct correlation with eGFR (Table 2). Additionally, plasma concentrations of SAM (but not those of SAH or Hcy) showed a negative correlation with BMI (Table 2).

Diabetes mellitus was associated with higher plasma concentrations of SAM. Gender and smoking were not associated with the concentrations of any of the three metabolites in plasma (Appendix A).

Participants who reached the predefined endpoint had significantly higher plasma concentrations of Hcy, SAH and SAM at baseline compared to those who did not (Figure 1A–C).

### 3.3. Associations Between C1 Metabolites and Cardiovascular Outcomes in Longitudinal Analyses

We next analyzed whether plasma concentrations of C1 metabolites predict cardiovascular events in CKD patients. During a median (IQR) follow-up period of 4.0 [3.2; 4.3] years, 55 participants experienced MACE.

Kaplan–Meier survival analysis showed that higher plasma concentrations of SAH (Figure 2), SAM (Figure 3) and Hcy (Figure 4) were associated with the risk of MACE (*p* < 0.001, *p* = 0.01 and *p* = 0.008, respectively). Results of Kaplan–Meier survival analysis after stratification by smoking and diabetes status are shown in the Appendix A.

The HR (95% CI) of MACE for the third tertiles of plasma concentrations of SAH [3.544 (1.751–7.935)] and SAM [3.073 (1.474–7.017)] were significant in the univariate analyses. Adjustments for renal function markers and traditional cardiovascular risk factors abolished the associations between SAH and SAM and MACE. The HR (95% CI) of MACE for the upper tertile of plasma Hcy compared to the lowest tertile were significant only in the univariate analysis (Table 3 and Figure 5).

## 4. Discussion

We investigated the association between plasma concentrations of SAH, SAM and Hcy and the future risk of MACE in patients with CKD. Our study has shown that, although the associations were significant in the univariate analyses, adjustments for traditional risk factors abolished these associations, suggesting that SAH, SAM and Hcy are not independent risk factors or risk markers for MACE among patients with CKD (KDIGO GFR category G2 to G4).

Epidemiological studies reported significant associations between plasma concentrations of Hcy and cardiovascular diseases. However, randomized controlled trials using high-dose folate, vitamin B6 and vitamin B12 to lower Hcy levels failed to reduce cardiovascular event rates [13]. Most of these trials focused on Hcy and did not investigate plasma concentrations of SAH and SAM. This is of special interest, as vitamin supplementation that lowers plasma Hcy does not lower plasma concentrations of SAH and SAM [14], suggesting these two metabolites, rather than Hcy, are cardiovascular risk factors [9].

In healthy subjects, SAH is either hydrolyzed back into Hcy and adenosine or excreted by the kidneys. A previous clinical trial demonstrated that 40% of extracellular SAH is extracted by the kidney [7]. Our results confirm moderate-to-strong associations between kidney function markers and concentrations of SAH, SAM and those of Hcy. The inverse correlation between SAH and eGFR has been reported even among healthy people with normal kidney function (I Like HOMe) [15]: In line with the present study among CKD patients, also among healthy subjects, the strongest correlation between C1 metabolites and eGFR was found for SAH (SAH: r = −0.335, *p* < 0.001; SAM: r = −0.157, *p* = 0.002; Hcy: r = −0.250, *p* < 0.001). Moreover, when comparing the mean plasma levels of the C1 metabolites between the two studies, the difference was strongest for SAH (SAH: 4.4 ×, SAM: 1.9 ×, Hcy: 1.6 × times higher levels in CARE FOR HOMe than in I Like HOMe) [15].

This observation suggests that the association of elevated SAH and worse cardiovascular outcome is confounded by renal function [16]. The accumulation of SAH may cause inhibition of DNA-methyltransferases and therefore, epigenetic dysregulation. Several functional genes that have been related to atherosclerosis are regulated by methylation, suggesting that elevated concentrations of SAH due to kidney dysfunction may contribute to the increased cardiovascular morbidity of patients with CKD [4,6]. Furthermore, given that SAH is a by-product of the endogenous nitric oxide synthase inhibitor asymmetric dimethylarginine (ADMA), SAH accumulation may promote atherosclerosis via the ADMA pathway [17].

In line with the present results, we have previously shown that SAH, but not Hcy, is associated with traditional cardiovascular risk factors and subclinical arteriosclerosis (intima-media thickness) among 402 subjects with low cardiovascular risk and with normal kidney function (mean eGFR = 101 mL/min/1.73 m^2^) [15].

Longitudinal studies on SAH and SAM as cardiovascular predictive or prognostic markers are sparse. Xiao et al. reported a significant association between higher plasma SAH levels and the risk of cardiovascular events among 1003 people undergoing coronary angiography [10]. In their analysis, Hcy was not associated with cardiovascular events. In contrast to our study, the participants in the study of Xiao et al. had normal kidney function (creatinine at baseline = 82.0 µmol/L), and the median values of SAH, SAM and Hcy were lower than in our study [10]. In our study, SAH was not associated with MACE after adjustment for eGFR and albuminuria. We found similar results for SAM; again, higher SAM levels were associated with the occurrence of cardiovascular events in univariate analysis but not after adjustment for eGFR and albuminuria. In contrast, a recently published study revealed an inverse association between SAM levels and risk of mortality in patients with known coronary artery disease and normal kidney function [18]. In this study, levels of all three C1 metabolites were notably lower than in our present analysis among CKD patients. In a subgroup analysis considering only participants with GFR < 60 mL/min/1.73 m^2^, SAM was no longer associated with the predefined outcome [18].

Our study has several strengths: to the best of our knowledge, our present analysis is the largest study in CKD patients focusing on the association between parameters of the C1 metabolism, kidney function and cardiovascular outcome. Moreover, the CARE FOR HOMe FU cohort included patients with a wide range of eGFR and has a follow up over four years. As a limitation, levels of B vitamins were not measured, which may have provided additional information in this context. In addition, we did not include people with normal eGFR, suggesting that the results cannot be generalized to a non-CKD population. Furthermore, this study is an observational design, and thus the results might not be appropriate for causal inference.

## 5. Conclusions

In summary, our study provides novel data on the association between parameters of C1 metabolism and cardiovascular outcomes in CKD. Neither Hcy nor SAH or SAM were independently associated with cardiovascular outcome after correcting for confounders, especially renal function markers. Disturbed renal function may explain elevated C1 metabolites [19] and disturbed transmethylation. Future studies may investigate whether interventions to reduce SAH may modify the risk of cardiovascular events in people with CKD.

## Figures and Tables

**Figure 1 nutrients-17-00626-f001:**
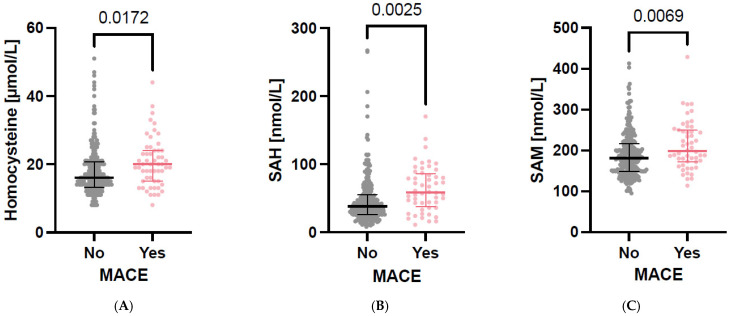
(**A**–**C**). Participants with MACE (major cardiovascular events) had significantly higher levels of plasma homocysteine (*p* = 0.0172), plasma S-adenosylhomocysteine (SAH; *p* = 0.0025) and plasma S-adenosylmethionine (SAM; *p* = 0.0069) than participants without MACE. Data are shown as scatter plots.

**Figure 2 nutrients-17-00626-f002:**
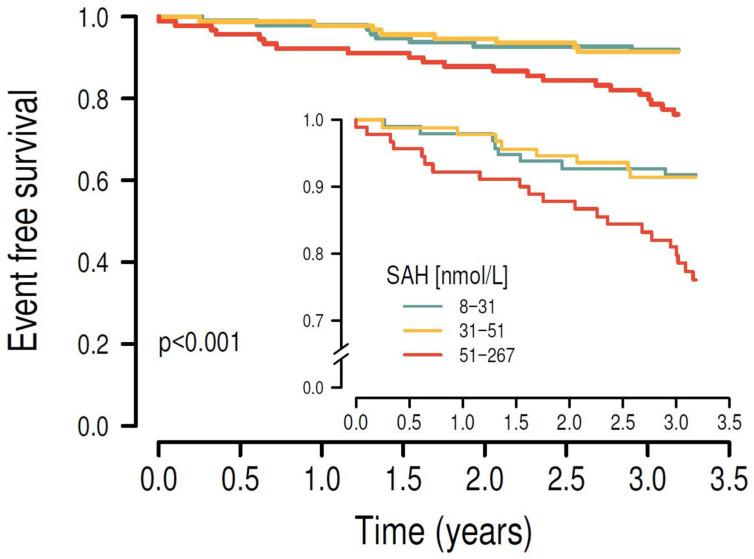
After stratifying the study cohort by their plasma SAH levels into tertiles, higher plasma SAH levels were significantly associated with the primary endpoint in univariate Kaplan–Meier survival analysis (*p* < 0.001).

**Figure 3 nutrients-17-00626-f003:**
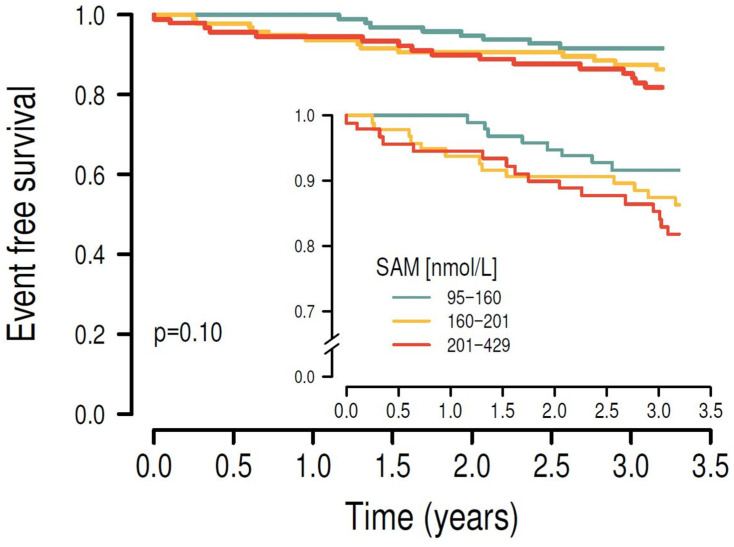
After stratifying the study cohort by their plasma SAM levels into tertiles, higher plasma SAM levels were significantly associated with the primary endpoint in univariate Kaplan–Meier survival analysis (*p* = 0.010).

**Figure 4 nutrients-17-00626-f004:**
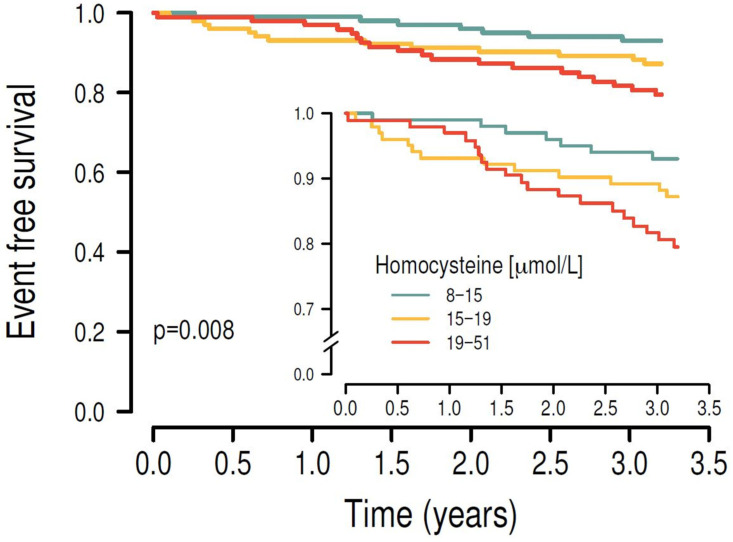
After stratifying the study cohort by their plasma homocysteine levels into tertiles, higher plasma homocysteine levels were significantly associated with the primary endpoint in univariate Kaplan–Meier survival analysis (*p* = 0.008).

**Figure 5 nutrients-17-00626-f005:**
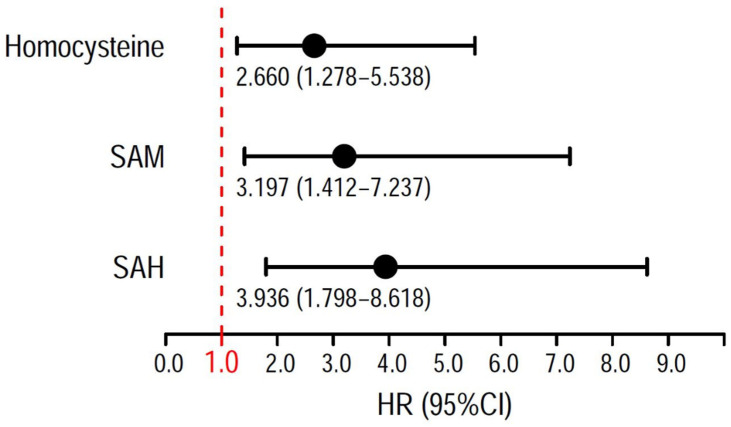
Forest plot: In univariate analysis, S-adenosylhomocysteine (SAH), S-adenosylmethionine (SAM) and homocysteine are significantly associated with the primary outcome.

**Table 1 nutrients-17-00626-t001:** Baseline characteristics.

	Total [n = 297]	CKD G2 [n = 58]	CKD G3a [n = 92]	CKD G3b [n = 79]	CKD G4/5 ND [n = 60]	CKD G5D [n = 8]	*p*
Patient demographics							
Age [years]	69.1 [60.4; 76.8]	60.8 [50.5; 67.9]	67.2 [58.8; 74.3]	75.7 [65.9; 79.8]	75.6 [69.1; 80.0]	63.5 [53.3; 77.2]	**<0.001**
Gender [female]	117 [39.4%]	23 [39.7%]	31 [33.7%]	36 [45.6%]	25 [41.7%]	2 [25.0%]	0.503
Smoking [yes]	32 [10.8%]	12 [20.7%]	11 [12.0%]	4 [5.1%]	4 [6.7%]	1 [12.5%]	**0.044**
Family’s history of CVD [yes]	61 [20.5%]	12 [20.7%]	21 [22.8%]	15 [19.0%]	8 [13.3%]	5 [62.5%]	**0.027**
CVD [prevalent]	95 [32.0%]	10 [17.2%]	25 [27.2%]	32 [40.5%]	23 [38.3%]	5 [62.5%]	**0.007**
Diabetes mellitus [yes]	106 [35.7%]	20 [34.5%]	30 [32.6%]	28 [35.4%]	25 [41.7%]	3 [37.5%]	0.850
BMI [kg/m^2^]	30 [27; 34]	30 [28; 33]	30 [26; 33]	30 [28; 34]	30 [27; 34]	28 [25; 35]	0.606
SBP [mmHg]	144 [132; 159]	139 [129; 155]	142 [132; 159]	144 [133; 155]	153 [133; 166]	152 [125; 181]	0.085
DBP [mmHg]	84 [76; 91]	86 [79; 94]	87 [79; 92]	80 [72; 89]	83 [75; 88]	82 [75; 84]	**<0.001**
Laboratory parameters							
Total Cholesterol [mg/dL]	179.8 ± 41.6	186.3 ± 36.4	185.4 ± 44.2	170.1 ± 39.0	177.6 ± 44.0	*	0.054
LDL-C [mg/dL]	102.7 ± 35.5	106.3 ± 32.1	107.3 ± 38.0	95.8 ± 31.7	101.4 ± 38.4	*	0.143
HDL-C [mg/dL]	53.2 ± 16.4	55.3 ± 15.8	55.8 ± 18.9	51.6 ± 15.8	49.3 ± 12.7	*	0.013
Triglyceride [mg/dL]	157.6 ± 84.2	167.4 ± 104.7	146.3 ± 79.9	150.7 ± 74.6	174.6 ± 78.6	*	0.532
CRP [mg/L]	2.5 [1.2; 5.5]	1.9 [1.1; 3.8]	2.6 [1.2; 6.0]	2.8 [1.3; 5.4]	3.0 [1.4; 6.0]	*	0.632
Renal parameters							
eGFR [mL/min/1.73 m^2^]	45 [32; 57]	68 [62; 75]	51 [47; 55]	38 [33; 40]	23 [19; 25]	*	**<0.001**
Plasma creatinine [mg/dL]	1.64 ± 0.72	1.04 ± 0.15	1.32 ± 0.21	1.62 ± 0.28	2.72 ± 0.78	*	**<0.001**
Albuminuria [mg/g crea]	38 [10;198]	18 [7; 110]	20 [8; 120]	39 [11;180]	124 [36; 508]	*	**0.001**
Phosphate [mg/dL]	3.36 ± 0.67	3.07 ± 0.58	3.18 ± 0.58	3.38 ± 0.58	3.90 ± 0.70	*	**<0.001**
One-carbon metabolites							
SAH [nmol/L]	41.5 [26.6; 63.9]	21.2 [16.1; 31.1]	32.2 [26.3; 42.0]	45.7 [40.4; 61.7]	82.3 [68.7; 100.3]	113.5 [92.2; 184.7]	**<0.001**
SAM [nmol/L]	183.4 [151.1; 223.5]	146.7 [129.7; 174.0]	165.4 [148.8 184.3]	197.4 [176.7; 222.7]	242.9 [205.4; 276.5]	291.1 [262.7; 317.2]	**<0.001**
Homocysteine [µmol/L]	16.6 [13.5; 21.2]	12.9 [11.3; 15.8]	15.7 [13.6; 18.0]	18.3 [15.3; 22.9]	22.4 [19.6; 29.4]	21.0 [16.4; 25.5]	**<0.001**

Data are shown as mean ± SD, median [IQR; 25th/75th percentiles], or n (%). CKD G2–G5 ND: chronic kidney disease KDIGO GFR categories G2 to G5 non-dialysis; G5 HD: KDIGO GFR category 5 on dialysis; CVD: cardiovascular disease; BMI: body mass index; SBP: systolic blood pressure; DBP: diastolic blood pressure; CRP: C-reactive protein; HDL-C: high density lipoprotein cholesterol; LDL-C: low density lipoprotein cholesterol; eGFR: estimated glomerular filtration rate according to the 2009 CKD-EPI creatinine-based equation; SAH: S-adenosylhomocysteine; SAM: S-adenosylmethionine. * No data available. Categorical variables are presented as absolute numbers and percentage and compared using Fisher’s test and chi-square test. Normally distributed variables are presented as mean ± standard deviation (SD) or as median [IQR; 25th/75th percentiles] in case of skewed distribution and compared by one-way ANOVA test. Significant values are given in bold letters.

**Table 2 nutrients-17-00626-t002:** Correlation between one-carbon metabolites and cardiovascular risk factors.

	SAH	SAM	Homocysteine	SAM/SAH
age	**0.263 *****	**0.191 *****	**0.285 *****	**−0.221 *****
BMI	−0.036	**−0.165 *****	−0.020	0.048
SBP	0.043	0.023	0.046	−0.041
Total Cholesterol	**−0.120 ****	**−0.126 ****	**−0.111 ****	0.074
Albuminuria	**0.250 *****	**0.208 *****	**0.110 ****	**−0.210 *****
eGFR CKD-EPI_crea_	**−0.588 *****	**−0.497 *****	**−0.424 *****	**0.442 *****

** *p* < 0.01; *** *p* < 0.001; BMI: body mass index; SBP: systolic blood pressure; eGFR CKD-EPI: estimated glomerular filtration rate calculated by CKD-EPI based on serum creatinine. The correlation coefficients between various continuous variables were calculated with Spearman test. Correlation coefficients with *p*-values below 0.05 are given in bold letters. SAM/SAH: quotient of both C1 metabolites.

**Table 3 nutrients-17-00626-t003:** Cox regression models (end point: MACE and all-cause mortality).

	Crude Model		Model 2		Model 3		Model 4	
Exposure Variable	HR (95% CI)	*p*-Value	HR (95% CI)	*p*-Value	HR (95% CI)	*p*-Value	HR (95% CI)	*p*-Value
Categories								
SAH								
First tertile (8–31 nmol/L; n = 99)	*1*		*1*		*1*		*1*	
Second tertile α (31–51 nmol/L; n = 99)	1.286 (0.531–3.196)	0.577	1.074 (0.440–2.687)	0.874	0.7609 (0.291–2.018)	0.574	1.443 (1.012–2.056)	**0.042**
Third tertile α (51–267 nmol/L; n = 99)	3.544 (1.751–7.935)	**<0.001**	2.696 (1.293–6.170)	**0.001**	1.645 (0.654–4.411)	0.303	1.209 (0.780–1.865)	0.393
SAM								
First tertile (95–160 nmol/L; n = 99)	*1*		*1*		*1*		*1*	
Second tertile α (160–20 nmol/L; n = 99)	1.862 (0.846–4.380)	0.132	1.357 (0.602–3.251)	0.471	1.159 (0.497–2.855)	0.737	1.124 (0.477–2.788)	0.792
Third tertile α (201–429 nmol/L; n = 99)	3.073 (1.474–7.017)	**0.004**	2.769 (1.324–6.335)	**0.009**	1.920 (0.764–5.138)	0.176	1.604 (0.605–4.470)	0.351
Homocysteine								
First tertile (8–15 µmol/L; n = 99)	*1*		*1*		*1*		*1*	
Second tertile α (15–19 µmol/L; n = 99)	1.534 (0.689–3.503)	0.296	1.171 (0.515–2.735)	0.707	1.131 (0.499–2.630)	0.768	1.396 (0.593–3.379)	0.447
Third tertile α (19–51 µmol/L; n = 99)	2.76 (1.409–5.804)	**0.004**	1.804 (0.868–3.974)	0.124	1.115 (0.521–2.546)	0.786	1.212 (0.56–2.786)	0.635
Continuous predictors								
log SAH	7.379 (2.512–21.92)	**<0.001**	5.771 (1.797–18.66)	**0.003**	2.237 (0.351–14.37)	0.396	1.312 (0.204–8.446)	0.775
log SAM	40.59 (3.968–413.6)	**0.002**	34.73 (3.015–390.8)	**0.004**	7.126 (0.271–186.9)	0.239	2.935 (0.086–103.1)	0.552
log homocysteine	11.42 (2.238–55.69)	**0.002**	4.709 (0.759–27.19)	0.089	1.572 (0.243–10.18)	0.634	1.615 (0.244–10.78)	0.618

Model 1 is the crude model. Model 2 is adjusted for age and gender. Model 3 is additionally adjusted for eGFR and log-transformed albuminuria, and Model 4 is adjusted for the same variables as in model 3 and additionally for active smoking, diabetes mellitus and prevalent cardiovascular disease. Collinearity between confounders were assessed by analyzing variance inflation factor (VIF). VIF > 10 was considered to indicate serious collinearity, thereby inducing unstable cox regression coefficients with large confidence intervals. Variable with high collinearity were excluded. MACE = major cardiovascular event, defined as carotid, coronary and peripheral arterial revascularization, stroke, acute myocardial infarction, major amputation, cardiovascular death; SAH = S-adenosylhomocysteine, SAM = S-adenosylmethionine. α Reference is the first tertile. Significant values are given in bold letters.

## Data Availability

The datasets generated and/or analyzed during the current study are not publicly available due to organizational reasons, but they are available from the corresponding author on reasonable request.

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
