# Peer review of "Association of Homocysteine, S-Adenosylhomocysteine and S-Adenosylmethionine with Cardiovascular Events in Chronic Kidney Disease"

_nutrients, 2025, doi:10.3390/nu17040626_

Round 1

Reviewer 1 Report

Comments and Suggestions for Authors

Overview

This study explores the prognostic value of Hcy, SAH, and SAM in patients with CKD through clinical data. However, the design of the article seems to lack a control group, the results are also negative, no innovative findings are made, the statistical selection is incoherent, and the results are presented incompletely. We believe that the article is not complete enough for publication.

Details

1. Clinical trials need to provide ethical approval documents or ethical approval numbers and conduct clinical registration.

2. Partial baseline data expression in clinical studies is not standardized. For example, age, sex, duration, WHR, BMI, eGFR, urine protein, and blood pressure are not normally distributed and cannot be represented by means and standard deviations. 

3. In addition to the baseline data for inter-group comparison and regression analysis, the authors should add correlation analysis in the middle to preliminarily count their associations.

4. The article does not appear to include normal controls, such as patients of G1 stage with normal renal function and proteinuria, or volunteers at normal physical examination centers, which is key to distinguishing between G2 and normal control populations. We think it seems that the author should elaborate on this issue.

5. Figure 1 does not reflect the transparency of the data, and we believe that scatter plots or violin plots should be used to show the distribution of the data.

6. There is no logic in the presentation of the relevant indicators in Table 1, for example, blood lipids should be grouped together instead of scattered in various positions all around the table.

7. The inclusion and exclusion criteria are not clear enough. The diagnostic criteria should be explicitly quoted, and the exclusion criteria specified, such as recent trauma, surgery, or tumor.

8. The conclusion of the article seems to be a negative result, so what is the research intention and new findings in this negative result? This seems to be an important question to judge whether an article is worthy of publication.

Comments on the Quality of English Language

The English could be improved to more clearly express the research.

Author Response

  1. Clinical trials need to provide ethical approval documents or ethical approval numbers and conduct clinical registration.

CARE FOR HOMe was initiated two decades ago, before cohort studies were generally registered on academic platforms. The ethical approval number is given in the revised manuscript. If necessary, we can provide a copy of the original ethical approval documents to the editorials’ office.

  1. Partial baseline data expression in clinical studies is not standardized. For example, age, sex, duration, WHR, BMI, eGFR, urine protein, and blood pressure are not normally distributed and cannot be represented by means and standard deviations. 

We revised and re-sorted table 1 according to Reviewer 1’s proposal. As Reviewer 1 prefers, we provide the parameters “age, BMI, eGFR, albuminuria, blood pressure, homocysteine, SAH and SAM” as median and interquartile range. As sex is a categorical variable, we presented it as absolute numbers and percentage.

  1. In addition to the baseline data for inter-group comparison and regression analysis, the authors should add correlation analysis in the middle to preliminarily count their associations.

According to Reviewer 1’s recommendations, we redefine table S1 to table 2 to present correlation analysis directly in the main text body. Furthermore, we have added the p-values directly to the baseline table for better readability.

  1. The article does not appear to include normal controls, such as patients of G1 stage with normal renal function and proteinuria, or volunteers at normal physical examination centers, which is key to distinguishing between G2 and normal control populations. We think it seems that the author should elaborate on this issue.

In a previous study (I Like HOMe), we already investigated levels of homocysteine (HCY), S-adenosylhomocysteine (SAH) and S-adenosylmethionine (SAM) among 420 apparently healthy subjects. Here, mean plasma HCY, SAH and SAM concentrations were 11.6 ± 3.7 µmol/l (1.6 x lower than in the present study among CKD patients), 11.5 nmol/l ± 3.8 nmol/l (4.4 x lower than in the present study among CKD patients) and 101.9 ± 30.4 nmol/l (1.9 x lower than in the present study among CKD patients), respectively, and correlated with the eGFR (HCY: r=-0.250, p<0.001; SAH: r=-0.335, p<0.001; SAM: r=-0.157, p=0.002). This is in line with our present study among patients with CKD, showing an increase of the levels of C1 parameters from CKD G2 to CKD G5D and the strongest correlation of SAH with the kidney function (HCY: r=-0.424, p<0.001; SAH: r=-0.588, p<0.001; SAM: r=-0.497, p<0.001). Moreover, I Like HOMe FU study found also a correlation between SAH and traditional cardiovascular risk factors as well as common carotid intima-media-thickness as a marker of subclinical atherosclerosis (r=0.129, p=0.010).

Following the Reviewer’s suggestion, we elaborated on the discussion and the limitations’ section with healthy subjects in our revised manuscript.

  1. Figure 1 does not reflect the transparency of the data, and we believe that scatter plots or violin plots should be used to show the distribution of the data.

Following Reviewer 1’s recommendations, we provide the data as scatter plots.

  1. There is no logic in the presentation of the relevant indicators in Table 1, for example, blood lipids should be grouped together instead of scattered in various positions all around the table.

We thank Reviewer 1 for his thoughtful recommendations and re-sorted and revised Table 1 according to this proposal.

  1. The inclusion and exclusion criteria are not clear enough. The diagnostic criteria should be explicitly quoted, and the exclusion criteria specified, such as recent trauma, surgery, or tumor.

We added detailed inclusion and exclusion criteria in the supplement of our manuscript.

  1. The conclusion of the article seems to be a negative result, so what is the research intention and new findings in this negative result? This seems to be an important question to judge whether an article is worthy of publication.

We generally feel that a publication of so-called “negative results” is as important as a publication of “positive results”. If the academic world only accepts papers with positive results, the pre-existing reporting bias would be become more pronounced

Reviewer 2 Report

Comments and Suggestions for Authors

The authors presented a relevant study of association of homocysteine, SAM and SAH with cardiovascular events in stage 2-5 CKD patients. The study needs to categorize the risk of MACE as per the different CKD stages such as risk with stage 2, risk with stage 3, risk with stage 4 and risk with stage 5 +/- dialysis to better understand the importance of these variables (homocysteine, SAM and SAH) with MACE. Many of these patients also had diabetes and were smokers- how do the authors explain the potential associations of these confounding risk factors with the MACE? were there patients with history of clotting or thromboembolic events? perhaps need to clarify that as well 

Author Response

  1. The study needs to categorize the risk of MACE as per the different CKD stages such as risk with stage 2, risk with stage 3, risk with stage 4 and risk with stage 5 +/- dialysis to better understand the importance of these variables (homocysteine, SAM and SAH) with MACE.

We thank Reviewer 2 for this recommendation and provide Kaplan Meier analysis for KDIGO category G2 to G5 and MACE to the point-to-point answer. If Reviewer 2 prefers, we can include the Kaplan Meier analysis to the Supplemental Data File.

  1. Many of these patients also had diabetes and were smokers- how do the authors explain the potential associations of these confounding risk factors with the MACE?

We have added outcome analyses separated for diabetes mellitus status and smoking status to the Supplemental Data File of our manuscript (Figure S1-S6).

  1. Were there patients with history of clotting or thromboembolic events?

Unfortunately, we have not predefined clotting or thromboembolic events as outcome parameters in CARE FOR HOMe.

Round 2

Reviewer 1 Report

Comments and Suggestions for Authors

Evaluation

The author answers some of our superficial questions, but avoids the important ones. For example, the ethics of clinical research and the nature of the study are not clear, and there are major data reliability problems in the statistical part of the current research methodology, and the confounding factors strongly affect the experimental results. Therefore, we believe that the article is not sufficient for publication in terms of ethics, research nature, and data reliability.

Details

Response to comment 1.

Our last review made it very clear to the first revision that clinical research needs to provide both ethical documents and clinical registration, and obviously the authors did not provide both. Most of the results of clinical data from almost 20 years ago are currently available, but the authors do not say whether they are only remembered now or collected 20 years ago. 

If it was collected 20 years ago, this study was a retrospective study, and it only looked for the records of the time, and it did not have any impact on the present, and we think that the methodology of the author's writing paper is like that of the current collection, which is misleading, and the entire methodology should be changed to a retrospective study, which will consume a lot of time and manpower. 

If the data collected are now collected from 20 years ago, then the authors should conduct clinical registration and provide ethical proof. 

Overall, in response to this opinion, we found that the authors were confused about the nature of the clinical study, and did not distinguish whether it was a retrospective study or a follow-up study 20 years later. This greatly affects the judgment of the nature of the article.

Response to comment 4.

The idea that the normal control group can be dispensed with this time is obviously logically wrong. Without the adjunct of the normal population, it is impossible to determine the universal applicability of the study, and there is no way to tell whether the study can be used for normal people or only patients.

Moreover, there was also a serious selection bias in statistically significant numbers without normal controls as an adjunct. We judged the current studies with no control group to have serious false positives and therefore the results are unreliable.

Response to comment 3 and 5.

The distribution in Figure 1 shows that the mean difference between groups in the case of a large sample is around 0.05, but the correlation between the two groups is very high, which is likely to be a confounding factor caused by correlation factors in other biochemical indicators other than the main factor. Subsequent regression analyses did not remove confounders, so we believe that the data are still false positives and that the current results are overstated.

Response to comment 8.

The authors argue that negative results are just as important as positive outcomes. However, the premise of this belief is that this result is fundamentally subversive or instructive for clinical practice with the previous scientific community understanding. Apparently, that issue is not explicitly stated in the article. Based on our responses above, we can also see that most of the statistics and even the nature of the article in this study are questionable, and we believe that it is not enough to talk about whether the article is important or not, but that the article should be authentic first. Thus, whether this negative result is reliable is also a question.

Comments on the Quality of English Language

The English could be improved to more clearly express the research.

Author Response

Point-to-point-answers

Reviewer 1:

“Response to comment 1: Our last review made it very clear to the first revision that clinical research needs to provide both ethical documents and clinical registration, and obviously the authors did not provide both. Most of the results of clinical data from almost 20 years ago are currently available, but the authors do not say whether they are only remembered now or collected 20 years ago. If it was collected 20 years ago, this study was a retrospective study, and it only looked for the records of the time, and it did not have any impact on the present, and we think that the methodology of the author's writing paper is like that of the current collection, which is misleading, and the entire methodology should be changed to a retrospective study, which will consume a lot of time and manpower. If the data collected are now collected from 20 years ago, then the authors should conduct clinical registration and provide ethical proof. Overall, in response to this opinion, we found that the authors were confused about the nature of the clinical study, and did not distinguish whether it was a retrospective study or a follow-up study 20 years later. This greatly affects the judgment of the nature of the article.”

  • We thank Reviewer 1 for his/her recommendation. The CARE FOR HOMe FU trial was approved by the local ethics committee “Ethikkommission der Ärztekammer des Saarlandes, Faktoreistraße 4, 66111 Saarbrücken, Germany,” on November 10th, 2011. The ethical code is 08/10. This information is provided in the manuscript body. If necessary, a certified translation of the approval documents can be provided to the Editorial Office.

Up to now, there was no registration of the study. But as Reviewer 1 proposed, we are inscribing the CARE FOR HOME trial retrospectively to the German Clinical Trials Register. Please note that it has not been fully enclosed by the day of our resubmission of the manuscript body.

“Response to comment 4: The idea that the normal control group can be dispensed with this time is obviously logically wrong. Without the adjunct of the normal population, it is impossible to determine the universal applicability of the study, and there is no way to tell whether the study can be used for normal people or only patients. Moreover, there was also a serious selection bias in statistically significant numbers without normal controls as an adjunct. We judged the current studies with no control group to have serious false positives and therefore the results are unreliable.”

  • The CARE FOR HOMe study was initiated to evaluate renal and cardiovascular risk factors of the predefined cohort of participants suffering from chronic kidney disease. No control group has been intended. The biomarkers S-adenosylhomocysteine, S-adenosylmethionine, and homocysteine were analyzed in this cohort (CARE FOR HOME FU study) and in another study of our group, the “I LIKE HOMe” study. This study included participants with normal kidney function and low cardiovascular risk. These results of the analyses are cited in our manuscript and had been published as a full manuscript before (Zawada et al, Atherosclerosis 234 (2014) 17-22).

“Response to comments 3 and 5: The distribution in Figure 1 shows that the mean difference between groups in the case of a large sample is around 0.05, but the correlation between the two groups is very high, which is likely to be a confounding factor caused by correlation factors in other biochemical indicators other than the main factor. Subsequent regression analyses did not remove confounders, so we believe that the data are still false positives and that the current results are overstated.”

  • We thank Reviewer 1 for these thoughtful comments and revised our results’ section. We mainly focused on the two groups, “MACE” and “No MACE” (Figure 1, Table S1) and reduced the number of confounders in our regression analysis after checking for collinearity.

“Response to comment 8: The authors argue that negative results are just as important as positive outcomes. However, the premise of this belief is that this result is fundamentally subversive or instructive for clinical practice with the previous scientific community understanding. Apparently, that issue is not explicitly stated in the article. Based on our responses above, we can also see that most of the statistics and even the nature of the article in this study are questionable, and we believe that it is not enough to talk about whether the article is important or not, but that the article should be authentic first. Thus, whether this negative result is reliable is also a question.”

  • We respect Reviewer 1’s evaluation, but we do not share his thoughts.

Reviewer 2 Report

Comments and Suggestions for Authors

The authors have satisfactorily revised the manuscript as recommended. 

Author Response

We are delighted that Reviewer 2 is satisfied with the revised version of our manuscript and we would like to thank him/her for his/her input and time.